# Microcapsule Preparation and Properties of Flavonoid Extract from Immature *Citrus reticulata* ‘Chachiensis’ Peel

**DOI:** 10.3390/foods13193096

**Published:** 2024-09-27

**Authors:** Xinyi Zhang, Qili Li, Sisi Wu, Yan Liu, Jiaxu Chen, Tao Li, Donglin Su

**Affiliations:** 1Longping Branch, Graduate School of Hunan University, Changsha 410125, China; zhangxinyi@hunaas.cn (X.Z.); wusisi@hunaas.cn (S.W.); liuyan@hunaas.cn (Y.L.); 2Hunan Agricultural Product Processing Institute, Hunan Academy of Agricultural Sciences, Changsha 410125, China; liqili@hunaas.cn; 3College of Food Science and Nutritional Engineering, China Agricultural University, Beijing 100083, China; chenjiaxu@hunaas.cn

**Keywords:** *Citrus reticulata* ‘Chachiensis’, plant-based protein, encapsulation, microcapsule

## Abstract

*Citrus reticulata* ‘Chachiensis’ is a citrus cultivar in the Rutaceae family, and its peel is commonly utilized as a raw material for Guangchenpi. This study used flavonoid extract from the peel of immature *Citrus reticulata* ‘Chachiensis’ (CCE) as the raw material to investigate the encapsulation ability of different wall materials (plant-based proteins, including soybean protein isolation (SPI), pea protein (PP), and zein; carbohydrates, including maltodextrin (MD), Momordica charantia polysaccharide (MCP), and gum acacia (GA); and composite wall materials of both types) on CCE. The wall material with the highest encapsulation rate was selected for the preparation of CCE microcapsules. Furthermore, the physicochemical characteristics, antioxidant capacity, bioavailability, and storage stability of the CCE microcapsules were explored. The results indicated that among all wall materials, the composite wall material PPMD had the highest encapsulation rate, which was 84.44 ± 0.34%. After encapsulation, the microcapsules tended to have a yellow color and exhibited characteristics such as system stability, low moisture content, and low hygroscopicity. In vitro antioxidant assays revealed that the encapsulation of CCE significantly increased the scavenging rates of DPPH and ABTS free radicals. In vitro gastrointestinal digestion experiments indicated that the release rate of PPMD-CCE in intestinal fluid was significantly greater than that of free CCE, ultimately reaching 85.89 ± 1.53%. Storage experiments demonstrated that after 45 days under various temperature and light conditions, the retention rate of CCE in the microcapsules was significantly greater than that of free CCE. The above findings provide new possibilities for the application of PP and plant proteins and lay a foundation for the future industrial application of CCE.

## 1. Introduction

The most widely-consumed fruit worldwide is citrus. The Food and Agriculture Organization (FAO) reported that 158.5 million tons of citrus were produced worldwide in 2022. The cultivation of *Citrus reticulata* cv. Chachiensis (CRC), a member of the Citrus genus and Rutaceae family, has a long history [1]. Because of its numerous biological properties, including anti-inflammatory, antiviral, antitumor, and hypolipidemic effects, it is widely employed in the food and medicinal industries [2]. Between 40% and 50% of the entire fruit mass of CRC is composed of the peel [3]. The Chinese Pharmacopoeia (2020 edition) [4] lists it as the official raw ingredient used in the creation of Guangchenpi. Flavonoids, volatile oils, limonoids, phenolic acids, alkaloids, and other bioactive compounds are present in CRC peel [5]. The most representative of these is flavonoids. According to previous research, the two most prevalent flavonoids in CRC peel were nobiletin and hesperidin, with levels of 16.91 ± 0.14 mg/g DW and 43.02 ± 0.37 mg/g DW, respectively [6]. In the Chinese pharmacopoeia, hesperidin, tangeretin, and nobiletin are used to regulate the quality of chenpi. Hesperidin, for example, must make up no less than 3.5% of dried chenpi. In dried Guangchenpi, the content of hesperidin must not be less than 2%, and the combined amount of tangeretin and nobiletin must not be less than 0.42%. Although flavonoids are beneficial to humans in many ways, they are also sensitive to environmental conditions such as pH, temperature, and light [7]. Furthermore, they have low permeability and solubility in the small intestine of humans, resulting in low bioavailability, which restricts their usefulness in the food, pharmaceutical, and other industries. Encapsulation is one of the most cutting-edge and promising technological processes for bioactive ingredient delivery devices [8], and it has emerged as a viable solution to many of the issues mentioned above.

Citrus flavonoid encapsulation wall components include vegetable gum, MD, casein, whey protein isolate, and other proteins and carbohydrates. Additionally, research has been conducted on the use of plant-based proteins as microcapsule wall materials since they are more ‘green’, more focused, and less expensive than animal proteins [9]. Because of their advantageous functional characteristics, plant-based proteins have been used extensively as partial substitutes for animal proteins in dietary applications [10]. They are still infrequently utilized, nevertheless, as a wall material to enclose NOB. To prevent drug-loaded NPs from resolving after oral delivery, Wu et al. [11] constructed nanoparticles designed from zein, tannic acid (TA), and nobiletin (NOB). They also presented a secondary protection mechanism. The preparation of naringenin–zein–sodium caseinate galactosylated chitosan nanoparticles (GC-NPs) for hepatocyte-specific targeting and their successful application in lipid-lowering therapy was demonstrated by Zhang et al. [12].

The raw material used in this study was immature *Citrus reticulata* ‘Chachiensis’ peel flavonoid extract (CCE). This study examined the effects of various wall materials, including plant-based proteins such as soybean protein isolate (SPI), pea protein (PP), and zein, and carbohydrate wall materials such as maltodextrin (MD), Momordica charantia polysaccharides (MCPs), and gum acacia (GA), and their composite wall materials, on the CCE. The index of encapsulation efficiency (EE) was used to choose the finest wall materials. Following process optimization, the antioxidant capacity in vitro, gastrointestinal digest capacity in vitro, and storage stability under various light and temperature conditions were measured for the PP-CCE, MD-CCE, and PPMD-CCE microcapsules. The objectives of this study are to develop a novel approach for the delivery of flavonoids in the functional food and pharmaceutical industries, increase their bioavailability, and provide a cost-effective, efficient, and ecologically friendly method.

## 2. Materials, Instruments, and Methods

### 2.1. Materials and Instruments

CCE, Lianyuan Kang Biotech Co., Ltd. (Lianyuan, China); Rutin analytical standard (≥97%), Shanghai Yuanye Biotechnology Co., Ltd. (Shanghai, China); SPI (food-grade model 100 gel type), Linyi Shansong Biological Products Co., Ltd. (Linyi, China); PP (food grade), Ebolai Food Ingredients Co., Ltd. (Guangzhou, Guangdong, China); Zein (food grade), Guangdong Runze Food Ingredients Co., Ltd. (Heyuan, Guangdong, China); MD DE12, Longrun food; Bitter Melon Polysaccharide (40%), domestic; Ethanol (≥99.7%); NaNO_2_ (analytical pure); Al (NO_3_)_3_ (analytically pure); NaOH (analytical purity); artificial gastric juice (the primary ingredients are pepsin, diluted acid, and sodium chloride), Leagene Biotechnology & Limited Inc. (Beijing, China); artificial intestinal fluid (the primary ingredients are trypsin and phosphate), Leagene Biotechnology Co., Ltd. (Beijing, China); DPPH Kit, Suzhou Comin Biotechnology Co., Ltd. (Suzhou, China); ABTS kit, Suzhou Comin Biotechnology Co., Ltd. (Suzhou, China)

Freeze-dryer, Model: LGJ-25C, Sihuan FricCO Technology Development (Beijing) Co., Ltd. (Beijing, China); Medical centrifuge, Model: DL-2018HR, Anhui Zhongke During Commercial Electric Co., Ltd. (Hefei, Anhui, China); Electronic Balance, Model: FA2204, Shanghai Puchun Measuring Instrument Co., Ltd. (Shanghai, China); Collector type constant temperature heating magnetic stirrer, Model: DF-101S, Shanghai Lichen Bangxi Instrument Technology Co., Ltd. (Shanghai, China); High speed disperser, Model: XHF-D, Ningbo Xinzhi Biotechnology Co., Ltd. (Ningbo, China); Water bath thermostatic oscillator, Model: SHA-B, Changzhou Aohua Instrument Co., Ltd. (Changzhou, China)

### 2.2. UHPLC-OE-MS Detection Method and Conditions

Utilizing a Vanquish (Thermo Fisher Scientific, Waltham, MA, USA) ultra-high performance liquid chromatograph(Shanghai, China), chromatographic separation was carried out using a Phenomenex Kinetex C18 (2.1 mm × 50 mm, 2.6 μm) liquid chromatograph. Phase B was isopropyl alcohol-acetonitrile (1:1, *v*/*v*), while Phase A was aqueous and included 0.01% acetic acid. The sample tray was 4 °C, the sample volume was 2 μL, and the column temperature was 25 °C. Primary and secondary mass spectrum data were acquired using the Orbitrap Exploris 120 mass spectrometer. There are 50 Arb of sheath gas flow, 15 Arb of auxiliary gas flow, 320 °C capillary temperature, 1 Arb sweep gas, and 350 °C carburetor temperature. The collision energy is SNCE 20/30/40, the spray voltage is 3.8 kV in positive ion mode and −3.4 kV in negative ion mode, and the full MS resolution is 60,000. The MS/MS resolution is 15,000.

### 2.3. Preparation and Wall Material Screening of CCE Microcapsules

It is somewhat altered by citing Zhao et al.’s study [13]. The main process of preparing CCE microcapsules was shown in Figure 1. A total of 10% (*w*/*w*) of mixed carrier solution was obtained by dispersing PP, SPI, and zein in distilled water with MD, GA, and MCP at a mass ratio of 2:3 at 40 °C. For two hours, the solution was agitated at 650 rpm/min. Following the addition of CCE at a 1:9 core–wall mass ratio, it was agitated for two hours at 50 °C at 650 rpm/min. For one minute, a high-pressure homogenizer operating at 8400 rpm/min was utilized. Finally, it was freeze-dried and pre-frozen at −80 °C. Using the index of EE, the ideal wall material for creating microcapsules was discovered.

### 2.4. Determination of Flavonoid Content and EE of CCE Microcapsules

#### 2.4.1. Determination of Flavonoid Content

The determination of flavonoid content was conducted using the NaOH-Al(NO_3_)_3_ method. In total, 0.2 g of freeze-dried microcapsules were accurately weighed and 70% ethanol solution was added at a ratio of 1:30 g/mL (absolute ethanol was used for surface flavonoid content determination). This was then fully oscillated and centrifuged at 5000 r/min at 16 °C for 10 min. Afterward, 3 mL of supernatant was taken and diluted with 30% ethanol solution to 10 mL. A total of 0.7 mL of 5% NaNO_2_ was added, mixed, and left for 6 min. Then, 0.7 mL of 10% Al(NO_3_)_3_ was added, mixed, and left for 6 min. Next, 5 mL of 4% NaOH was added and titrated to 25 mL with 30% ethanol. After mixing the mixture and letting it stand for 15 min, the absorbance was measured at 510 nm. Finally, the standard curve was used to determine the flavonoid content.

#### 2.4.2. Determination of EE

The amount of flavonoids present in the microcapsules served as a proxy for EE. The mix of the wall and core materials can be efficiently reflected by EE. Equation (1) was used to compute EE with reference to [14]: total flavonoid content (TFC) and surface flavonoid content (SFC) were measured in microcapsules.
(1)$$\mathrm{EE}(\%)=\left(\frac{\mathrm{TFC}-\mathrm{SFC}}{\mathrm{TFC}}\right)\times 100$$

### 2.5. Single-Factor Experiments with PPMD-CCE

The mass ratio of CCE to total wall material was 1:9, the mass ratio of PP to MD was 5:6, the reaction period was 15 min, and the reaction temperature was 40 °C. These were the fundamental process parameters. The wall material mass ratios were 1:6, 1:2, 5:6, 6:5, and 2:1 when all other factors were held constant. The mass ratios of the core–wall were 1:3, 1:9, 1:15, 1:21, 1:27, and the reaction times were 5 min, 10 min, 15 min, 20 min, and 25 min. The single-factor experiment was conducted with the reaction temperature set at 20 °C, 30 °C, 40 °C, 50 °C, and 60 °C. Lastly, an analysis was conducted on how different parameters affected the EE of CCE microcapsules.

### 2.6. PPMD-CCE Response Surface Experiment

The single-factor experiment was used to determine the CCE reaction temperature, which came out to be 40 °C. The inquiry elements that were chosen were the mass ratio of PP to MD (A), the mass ratio of CCE to total wall material (B), and the CCE reaction time (C). To find the best microcapsule preparation method, a three-factor, three-level response surface optimization analysis experiment was created using the EE as the response value and the Box-Behnken center combination design principle. Table 1 displays the parameters and level design.

### 2.7. Characterization of CCE Microcapsules

#### 2.7.1. Measurement of Particle Size and Zeta Potential

A laser diffractometer was used to measure the zeta potential and particle size distribution of microcapsules, as reported by Jain et al. [15]. The microcapsules were distributed in ethanol prior to measurement, and the sample solution was diluted for analysis. The ethanol refractive index (RI) was 1.36, and the testing temperature was 25 °C. Two min were spent equilibrating the samples in the device before the data were read.

#### 2.7.2. Color Measurement

Use a colorimeter to measure the color. Before taking a measurement, the equipment needs to be calibrated using a whiteboard. Following the completion of the calibration, the microencapsulated powder was put in a transparent glass dish, and the measurement was finished. Three distinct surfaces of the sample were chosen at random to calculate the color parameters L*, a*, and b*.

#### 2.7.3. Moisture Content

The 2 g microencapsulated samples were baked at 70 °C until they reached a consistent weight (M_t_). Equation (2) was used to determine the moisture content (MC), with M_0_ being the microcapsule sample’s initial weight.
(2)$$\mathrm{MC}(\%)=\left(\frac{M_{0}-M_{t}}{M_{0}}\right)\times 100$$

#### 2.7.4. Water Solubility Index

See Mahdi et al. [16] for small adjustments. The 1 g microcapsule sample was mixed with distilled water, magnetically agitated for 30 min at room temperature, and then centrifuged for 10 min at 5000 rpm/min. After collecting the supernatant, it was dried at 105 °C in a petri dish that had been previously weighed. The water solubility index (WSI) was computed using Equation (3):(3)$$\mathrm{WSI}(\%)=\frac{\mathrm{The}\;\mathrm{weight}\;\mathrm{of}\;\mathrm{sample}\;\mathrm{in}\;\mathrm{the}\;\mathrm{supernatant}}{\mathrm{Weight}\;\mathrm{of}\;\mathrm{the}\;\mathrm{sample}\;\mathrm{in}\;\mathrm{the}\;\mathrm{solution}}\times 100$$

The precipitated samples after centrifugation were collected to facilitate the determination of the water absorption index (WAI), which was calculated according to Formula (4):(4)$$\mathrm{WAI}(\%)=\frac{\mathrm{The}\;\mathrm{weight}\;\mathrm{of}\;\mathrm{the}\;\mathrm{precipitated}\;\mathrm{sample}}{\mathrm{Weight}\;\mathrm{of}\;\mathrm{the}\;\mathrm{sample}\;\mathrm{in}\;\mathrm{the}\;\mathrm{solution}}\times 100$$

#### 2.7.5. Bulk Density

After weighing and pouring 2 g of the microcapsule sample into a graduated cylinder, the powder inside the microcapsules organically sank 20 times after repeated vibrations until the sample could no longer be squeezed. The experiment was repeated three times. The mass of the microcapsule per unit volume was computed. Equation (5) was utilized to compute the bulk density (BD).
(5)$$\mathrm{BD}\;(g/{\mathrm{cm}}^{-3})=\frac{\mathrm{CCE}\;\mathrm{microcapsule}\;\mathrm{sample}\;\mathrm{quality}}{\mathrm{CCE}\;\mathrm{microcapsule}\;\mathrm{sample}\;\mathrm{volume}}$$

#### 2.7.6. Liquidity

Weigh the 3 g microcapsule sample precisely so that it spills out of the funnel and accumulates on the circular plate. Do this three times. Measurements were made of the covering radius R (cm) and the powder accumulation height H (cm). Equation (6) was used to obtain the sample’s angle of repose, or θ:(6)$$\theta =\arctan\left(\frac{H}{R}\right)$$

#### 2.7.7. Hygroscopicity

See Hoskin et al.’s experimental protocol [17]. Microcapsule samples weighing 1 g were precisely measured. They were kept at room temperature in airtight containers filled with a saturated NaCl solution. After seven days, the samples were weighed. Equation (7) was used to determine the samples’ hygroscopicity; M_1_ represented the mass of the sample following hygroscopicity, and M_2_ represented the sample’s original mass.
(7)$$\mathrm{Hygroscopicity}/\%=\left(\frac{M_{1}-M_{2}}{M_{2}}\right)\times 100$$

#### 2.7.8. Scanning Electron Microscopy (SEM)

The morphologies of three different types of microcapsules, as well as CCE, PP, and MD, were examined using China’s Hitachi Corporation’s (Beijing, China) S-4800 scanning electron microscope. A small layer of gold was applied after the spray-dried powder was placed atop a brief cylindrical column. Digital pictures were taken at a 3 kV accelerating voltage. The working factor of SEM is 1000 times.

#### 2.7.9. Fourier Transform Infrared Spectroscopy (FTIR)

The American Nicolet Company’s (Madison, WI, USA) Nicolet6700 FTIR spectrometer was used to acquire the samples’ FTIR spectra. The spectrometer had a resolution of 4 cm^−1^, a signal-to-noise ratio of 50,000:1, a scanning time of 32, and a wavenumber range of 400–4000 cm^−1^.

#### 2.7.10. X-ray Diffraction (XRD)

For XRD analysis, the D8 Advance X-ray diffractometer from the German company Bruker (Berlin, Germany) was utilized, with a tube voltage of 40 kV and a current of 40 mA. A scan speed of 3 °/min was employed to measure the 2θ scattering angle between 10° and 99°, with a step size of 0.013.

#### 2.7.11. Differential Scanning Calorimetry (DSC)

Under the protection of a nitrogen environment (60 mL/min), 10 mg samples were heated on an aluminum tray from 30 °C to 400 °C at a rate of 10 °C per minute for each measurement.

### 2.8. Antioxidant Capacity of CCE Microcapsules

#### 2.8.1. DPPH Radical Scavenging

The CCE and microcapsule contents were 30 μg/mL, 60 μg/mL, 90 μg/mL, 120 μg/mL, and 150 μg/mL. Suzhou Comin Biotechnology Co., Ltd.’s (Suzhou, China) total antioxidant capacity (DPPH technique) kit was utilized to ascertain the measurement. The measurement of light absorption was conducted at 515 nm following a 20 min period of darkness. Formula (8) used the elimination of free radicals to express antioxidant capacity:(8)$$\mathrm{Radical}\;\mathrm{scavenging}\;\mathrm{capacity}\;(\%)=\left(1-\frac{\mathrm{As}}{\mathrm{Ac}}\right)\times 100$$

#### 2.8.2. ABTS Radical Scavenging

Microcapsules and CCE had concentrations of 20 μg/mL, 40 μg/mL, 60 μg/mL, 80 μg/mL, and 100 μg/mL. Suzhou Comin Biotechnology Co., Ltd.’s(Suzhou, China) total antioxidant capacity (ABTS technique) kit was utilized to ascertain the measurement. Within ten min of complete mixing, the absorbance value at 734 nm was determined. Formula (8), which measures the elimination of free radicals, expresses antioxidant ability.

### 2.9. The Simulated Gastrointestinal Digestion Experiment

This has been somewhat altered in accordance with Wang et al.’s experimental methodology [18]. Three microcapsules and 0.5 g of CCE were precisely weighed and placed in a centrifuge tube with 20 mL of synthetic gastric juice. Following four hours of shaking at 37 °C and 110 rpm in a shaker, the sample was centrifuged for 20 min at 4000 rpm. After decanting the supernatant, 20 mL of synthetic intestinal fluid was added. For six hours, the shaking procedure was maintained at 37 °C and 110 rpm. To measure the flavonol content throughout this period, 1 mL of digestive fluid was inhaled every hour. To maintain the same volume of solution in the centrifuge tube, the appropriate volume of digestion solution was supplemented at the same time. The CCE release rate was computed using Formula (9) as follows:(9)$$\mathrm{CCE}\;\mathrm{release}\;\mathrm{rate}(\%)=\left(\frac{\mathrm{Flavonoid}\;\mathrm{content}\;\mathrm{in}\;\mathrm{digestive}\;\mathrm{juices}}{\mathrm{Flavonoid}\;\mathrm{content}\;\mathrm{in}\;\mathrm{microcapsules}}\right)\times 100$$

### 2.10. Storage Stability of CCE Microcapsules

#### 2.10.1. Temperature Stability

For 45 days, CCE and three different types of microcapsules were split into three sections, sealed in a brown sample vial, and kept at 4 °C, 20 °C, and 55 °C, respectively. Every five days, the flavonoid concentration of 0.2 g samples was found. Retention efficiency was calculated using a slightly modified version of Reference [19]. Equation (10) for the calculation was as follows:(10)$$\mathrm{Retention}\;\mathrm{rate}(\%)=\left(\frac{\mathrm{Flavonoid}\;\mathrm{content}\;\mathrm{after}\;\mathrm{storage}}{\mathrm{Initial}\;\mathrm{flavonoid}\;\mathrm{content}}\right)\times 100$$

#### 2.10.2. Light Stability

The three microcapsules and the CCE were split into two halves. A brown sample bottle was used to store one piece in the dark, and a light-permeable sample bottle was used to store the other portion. It was kept for 45 days in natural light. Every five days, the flavonoid concentration of 0.2 g samples was found, in line with Formula (10).

### 2.11. Statistical Analysis

Every test was conducted three times. Software from IBM SPSS Statistics 26 was utilized to examine the significance of the difference (*p* < 0.05). The response surface experiment was designed and analyzed using Design-Expert 12. The mapping was performed using Origin 2021. Each and every value was given as the mean ± standard deviation.

## 3. Results and Discussion

### 3.1. UHPLC-OE-MS Detection Results

The total ion flow diagram (TIC) in positive and negative ion modes was obtained after CCE was examined via UHPLC-OE-MS in accordance with chromatographic and mass spectrometry conditions, as illustrated in Figure 2A. A total of 659 compounds were found and recognized via CCE after the data that had been received and identified via the BiotreeDB (V3.0) database were processed. Alkaloids, flavonoids, fatty acids and conjugates, phenolic acids, small peptides, phenylpropanoids, steroids, and saccharides were the primary constituents. Flavonoids made up the largest percentage of these compounds. Furthermore, the peak regions of the 104 flavonoids were compared, and Table 2 displays the compositions and proportions of the top 10 flavonoids. Hexamethylquercetagetin, 3′-hydroxy-7,8,4′,5′-tetramethoxyflavone, heptamethoxyflavone, hesperetin 7-neohesperidoside, and hesperidin were the five flavonoids with the greatest amounts, in that order. In CCE, they made up 79.7813%, 7.6861%, 2.2447%, 1.2732%, and 1.2732% of the flavonoids in that order.

### 3.2. Determination of the CCE Microcapsule Wall Material

Plant-based protein wall materials (SPI, PP, and zein), carbohydrate wall materials (MD, MCP, and GA), and composite wall materials made by combining the two materials were used to encapsulate CCEs under the same experimental circumstances. The outcomes are displayed in Table 3 and Figure 3, where the EE varies depending on the wall material used for encapsulation, ranging from 6.09 ± 0.03% to 82.79 ± 0.15%. The findings demonstrated that various wall materials had a substantial effect on the EE of CCEs. Compared with those of single wall materials such as PP-MD, PP-MCP, PP-GA, and zein-GA, the EE of the former was greater. PP-MD had the highest EE of all the wall materials (82.79 ± 0.15%). In addition, the EE was greater than that of the other microcapsules. In particular, the EE increased by 1260.16 ± 8.43% when zein was used as the wall material. Therefore, PP-MD was chosen as the wall material for preparing CCE microcapsules.

### 3.3. PPMD-CCE Microcapsule Preparation Process Optimization

#### 3.3.1. Single-Factor Experiment

The mass ratios of the wall material, core wall, reaction duration, and reaction temperature of the PPMD-CCE were all examined in this experiment. Figure 4A–D present the findings. The EE of PPMD-CCE reached a maximum when the mass ratio of PP to MD was 5:6, the CCE to total wall material ratio was 1:15, the reaction period was 10 min, and the reaction temperature was 40 °C. The encapsulation rate of PPMD-CCE was not significantly affected by the reaction temperature, and the EE reached 75.28 ± 1.60% or higher. As a result, the fixed condition for the trials was set at 40 °C.

#### 3.3.2. Response Surface Optimization Test

Table 4 displays the response surface experimental design and outcomes. Using software for multiple regression analysis, quadratic multinomial regression equations for the mass ratio of PP to MD (A), the mass ratio of CCE to total wall material (B), and the reaction time of CCE (C) of PPMD-CCE were found: Y = 83.31 − 2.32A + 0.8038B − 0.6563C − 3.15AB + 2.12AC + 0.3450BC − 3.48A2 − 5.69B2 − 3.93C2. Table 5 displays the regression equation’s variance analysis and significance test. For the PPMD-CCE EE, the independent variables A and B and the quadratic variables A2, B2, and C2 were highly significant. There was a considerable impact of the independent variable C on the PPMD-CCE EE. This suggests a direct correlation between these three parameters and the PPMD-CCE encapsulation rate.

The response surface graph immediately reflects each factor’s impact on the response value as well as the interaction between the two elements. Figure 4E–J display the fitting findings. Each of the three parameters had a substantial interaction with the others, and the order of influence on the EE of PPMD-CCE was as follows: Factor A (mass ratio of PP to MD) > Factor B (mass ratio of CCE to total wall material) > Factor C (reaction time of CCE).

#### 3.3.3. Optimization and Verification of the PPMD-CCE Preparation Process

The mass ratio of PP to MD was 0.646, the mass ratio of CCE to total wall material was 0.085, and the CCE reaction time was 8.943 min, according to software analysis, which determined the ideal encapsulation conditions for PPMD-CCE. The EE of PPMD-CCE was 83.98% in this scenario. The ideal procedure was altered in the following ways in light of the current circumstances: the mass ratio of CCE to total wall material was 0.08 (2:25), the mass ratio of PP to MD was 0.7 (7:10), and the reaction duration was 9 min. The verification test results under these circumstances revealed that the EE of PPMD-CCE was 84.44 ± 0.34%, which did not differ significantly from the expected value. This finding demonstrates that the quadratic regression model may be utilized to optimize the PPMD-CCE preparation process and has good accuracy and practicality.

### 3.4. The Physicochemical Properties of CCE Microcapsules

#### 3.4.1. Particle Size and the Zeta Potential

Table 6 displays the particle size, zeta potential, and polymer dispersion index (PDI) of CCE microcapsules made with various wall materials. The microcapsule size is immediately reflected in the size of the particles. There were notable differences in the particle sizes of the PP-CCE, MD-CCE, and PPMD-CCE microcapsules. PPMD-CCE had a particle size of 10.2733 ± 0.1966 μm, which was in the middle of those of PP-CCE and MD-CCE.

The particle size distribution is characterized by the PDI value, which is a crucial metric that typically ranges from 0 to 1. A PDI value of less than 1 indicates that the particles are physically stable, and a smaller PDI value indicates a more uniform particle size distribution [20,21]. The PDI values of PP-CCE, MD-CCE, and PPMD-CCE were 0.5371 ± 0.0133, 0.4583 ± 0.0220, and 0.5877 ± 0.0216, respectively. As a result, the particle size distributions of the three types of microcapsules are relatively uniform.

The zeta potential, which is related to the stability of particle aggregation, is a measure of the strength of the mutual repulsive force between particles. The results revealed that the zeta potential of PPMD-CCE was −55.1167 ± 0.7333 mV, which was significantly greater than the values of the other two microcapsules. The system has great dispersibility, stability, and repulsive force when the CCE is enclosed with two types of wall materials.

#### 3.4.2. Color

Table 7 shows the three different color parameters of the microcapsules and the free CCE. The brightness (L*), red/green (a*), and yellow/blue (b*) of the three microcapsules varied significantly from one another. PP-CCE and free CCE did not significantly differ in their a* values, whereas the a* values of the other two microcapsules did. PP-CCE, MD-CCE, and PPMD-CCE all had b* values that were noticeably greater than those of free CCE. The three different types of microcapsules tend to be yellow in hue because of the wall’s inherent tint.

#### 3.4.3. Other Physicochemical Properties

The three microcapsules were examined for MC, WSI, BD, liquidity, and hygroscopicity; Table 8 displays the experimental findings. The MCs of the three different types of microcapsules in this experiment were 6.9467 ± 0.2250% for PPMD-CCE, 4.8900 ± 0.8253% for MD-CCE, and 5.6167 ± 0.6167% for PP-CCE.

The ability of powder particles to dissolve in water is reflected in the WSI. The experimental results indicate that, at 14.80 ± 0.0800%, PP-CCE has the lowest solubility. On the other hand, MD-CCE had a far greater solubility, measuring 86.48 ± 0.7110%. The solubility of PPMD-CCE was 49.02 ± 0.0640%, which fell between the above two values.

The WAI is an important index for measuring water absorption. PP-CCE had the lowest WSI, but it had the greatest WAI, reaching 4.52 ± 0.0794%, according to the water absorption index. This could be a result of the superior water retention yet low solubility of PP. In conclusion, the combination of PP and MD can enhance this performance.

BD, which is correlated with the size of the space between powder particles, is a crucial measure for assessing the texture of microcapsules. The BDs of PP-CCE and MD-CCE did not significantly differ from one another. Their respective values were 0.3896 ± 0.0043 g/cm^−3^ and 0.3822 ± 0.0042 g/cm^−3^. The BD of PPMD-CCE, which was 0.3261 ± 0.0031 g/cm^−3^, differed significantly from the BDs of the first two microcapsules.

An essential metric for assessing the solubility of powdered microcapsule particles is the angle of repose. PP-CCE, MD-CCE, and PPMD-CCE had angles of repose that were less than 45°, specifically, 29.4784 ± 0.6910°, 39.6437 ± 0.7148°, and 38.3254 ± 0.5791°, respectively. The results demonstrate that all three microcapsules have comparatively good liquidity.

The findings of the hygroscopicity testing demonstrated that PP-CCE had a much greater hygroscopicity, reaching 15.4767 ± 0.0252%. The hygroscopicities of MD-CCE and PPMD-CCE, which are 15.0533 ± 0.0306% and 15.0900 ± 0.0300%, respectively, do not differ significantly, indicating that the two microcapsules are better suited for storage.

### 3.5. Characterization of the CCE and Its Microcapsules

#### 3.5.1. SEM

Figure 5 shows SEM images of the three microcapsules, CCE, PP, and MD. The images revealed that the MD had an uneven sheet shape, the PP was spherical, and the CCE was rod shaped. Nonetheless, no CCE morphology was observed, and the fundamental PP and MD morphologies were preserved in the encapsulated microcapsules. Therefore, the CCE was securely encased in the wall material. A comparison of the SEM images of the wall materials and the microcapsules clearly revealed that the latter had deeper and more numerous morphological depressions. This is in line with the findings of Sun et al. [22]. This effect is caused by the homogeneous loss of moisture during the freeze-drying process, which affects both the inside and the outside of the particles.

#### 3.5.2. FTIR Spectroscopy Analysis

The characteristic bands found in the FTIR spectra of CCE, PP, MD, and the three different types of microcapsules were linked to their molecular structures and chemical groups [23]. Figure 6A–F present the findings. PP is a plant-based protein with a normal protein structure. The locations of amide I, amide II, and amide III were 1628 cm^−1^, 1512 cm^−1^, and 1228 cm^−1^, respectively [24]. Furthermore, the PPMD-CCE still exhibited this structural trait. The MD results in a distinctive peak contraction vibration at 991 cm^−1^.

There were distinct bands in the PP-CCE, MD-CCE, and PPMD-CCE spectra between 3500 cm^−1^ and 3000 cm^−1^. This occurred because, to varying degrees, PP, MD, and PP-MD all combine with CCE to increase the quantity and intensity of the -OH absorption peaks. All three microcapsules showed a redshift or blueshift of the C=O peak at approximately 1642 cm^−1^, which suggests that the carbonyl group had stretched. Near 1361 cm^−1^ and 1072 cm^−1^, the stretching vibration of -CH_3_ and the antisymmetric stretching vibration peak of C-O-C were detected. Moreover, the spectrograms of the three microcapsules presented no distinguishable peaks other than those of CCE, PP, and MD. This demonstrated how the CCE is securely encased in wall material to preserve the integrity and natural state of the CCE.

#### 3.5.3. XRD Analysis

A popular technique for researching encapsulation systems made up of wall and core materials is X-ray diffraction (XRD). The diffraction patterns of various inclusion materials and core or wall materials may differ [25,26]. The X-ray diffraction patterns of three microcapsules, CCE, PP, and MD, are displayed in Figure 6G The crystal characteristics of the CCE are demonstrated by the several distinctive, sharply derived peaks observed in the diffraction pattern. Other flavonoids produced similar outcomes. For example, pure naringin was shown to be crystal-like by F. Sansone et al. [27], and quercetin powder was found to have crystal-like peaks by Hector Pool et al. [28]. Nevertheless, the X-ray diffraction patterns of the three microcapsules and the two wall materials did not show any identical characteristic crystal peaks; instead, the predominant feature was an amorphous structure. Consequently, CCE has been effectively contained in wall materials made of PP, MD, and PP-MD.

#### 3.5.4. DSC Analysis

To reflect the storage conditions of the CCE and microcapsules, DSC analysis takes into account the thermal stability of the substance. Figure 6H displays the outcomes of the experiments. Three different types of microcapsules, CCE, PP, and MD, all presented clear endothermic peaks between 50 °C and 165 °C, indicating denaturation and water loss. The endothermic peaks of the three microcapsules, which are wider than those of the CCEs, exhibit different phenomena. The reason may be the energy change generated by the electrostatic interaction between the core material and the wall materials. Furthermore, CCE curve analysis revealed a clear exothermic peak at 342 °C, which could be related to the breakdown of secondary metabolites in CCE [19]. The exothermic peaks of the three different types of microcapsules increased to 298 °C, 323 °C, and 310 °C in succession at the same time. The physical connection between the CCE and the wall materials, as well as the fact that the CCE was well encapsulated in the wall materials, were substantiated by these changes in thermal transitions [29]. The absence of the endothermic peak of CCE in the microcapsules lends further credence to this view. Through data analysis, the glass conversion temperatures of the three microcapsules were 38.85 °C, 43.81 °C, and 41.03 °C. At room temperature, the temperature is lower than the above temperature. Therefore, a stable glassy state may be maintained at room temperature (25 °C) [30], which makes the CCE stable and favorable for product quality and storage [31].

### 3.6. In Vitro Antioxidant Capacity of CCE and Its Microcapsules

Figure 7 displays the results of the antioxidant capacity experiments. Compared with free CCE, the three microcapsules demonstrated greater free radical scavenging activity at 30–150 μg/mL, and this ability improved as the concentration increased. At 150 μg/mL, PP-CCE had the strongest DPPH radical scavenging ability, reaching 68.19 ± 0.20%. The second highest percentage, 61.07 ± 0.34%, was associated with PPMD-CCE. Similarly, after CCE was encapsulated by three different types of wall materials, the ABTS free radical scavenging ability dramatically improved at 20–100 μg/mL. Among these compounds, PPMD-CCE outperformed the other compounds at all concentrations. At 100 μg/mL, the ABTS free radical scavenging ability of CCE was 73.75 ± 0.25%. Compared with free CCE, all three microcapsules ultimately demonstrated greater antioxidant activity. The findings of Hu et al. [32] further demonstrated that encapsulation is a successful strategy for increasing antioxidant capacity.

### 3.7. In Vitro Gastrointestinal Digestion and Release of CCE and Its Microcapsules

Figure 8 displays the outcomes of the experiment. Each component’s hourly flavonoid release rate is represented by A, and the change in the hourly release rate to final release rate ratio is represented by B. PPMD-CCE had the lowest release rate of the three microcapsules in gastric juice after four hours of in vitro digestion—just 19.75 ± 1.53%. PP-CCE and MD-CCE were released at 33.94 ± 1.76% and 32.10 ± 5.35%, respectively. After six hours of in vitro digestion, the release rates of PP-CCE and MD-CCE in the intestinal fluid were 77.64 ± 4.66% and 78.40 ± 5.35%, respectively. With a release rate of 85.89 ± 1.53%, PPMD-CCE demonstrated the highest bioaccessibility.

### 3.8. Storage Stability of CCE and Its Microcapsules

#### 3.8.1. Temperature Stability Experiment Results

Figure 9A–C display the flavonoid retention rates of three different types of microcapsules and free CCE after 45 days of storage at 4 °C, 20 °C, and 55 °C. There was a noticeable decline in the flavonol retention rate of free CCE when the storage time was between 30 and 45 days. At low temperature (4 °C), it decreased from 62.61 ± 3.13% to 44.68 ± 1.22%; at room temperature (20 °C), it decreased from 62.41 ± 1.51% to 35.82 ± 0.44%; and at high temperature (55 °C), it decreased from 61.19 ± 2.28% to 16.26 ± 0.15%. This suggests that when the temperature increases, the stability of the CCE deteriorates. For 45 days, the encapsulation of free CCE can enhance the retention of flavonoids, with the protective effect on CCE being more pronounced at lower temperatures. At all three temperatures, the flavonoid retention rate at 45 days was greatest for PPMD-CCE.

#### 3.8.2. Light Stability Experiment Results

For 45 days, storage studies were conducted in two types of light environments: natural light and artificial light. Figure 9D,E display the outcomes of the experiment. Free CCE decreased from 62.41 ± 1.51% to 37.04 ± 0.36% in the dark and from 60.71 ± 0.49% to 21.79 ± 0.34% under natural light over the course of 30 to 45 days of storage. It is evident that light significantly affects CCE storage stability. Following CCE encapsulation, flavonoids can be efficiently retained in natural light and protected from light for 45 days. PPMD-CCE outperforms the other two microcapsules in terms of light stability.

## 4. Discussion

Enhancing stability and bioavailability while shielding active components from the damaging effects of the environment is now possible with microencapsulation. The two most commonly used techniques are freeze drying and spray drying. For bioactive compounds that are thermosensitive, freeze-drying encapsulation is a more acceptable method as it can prevent thermal degradation and spray drying necessitates high temperatures [33]. Moreover, research has revealed that samples that have been freeze-dried have better retention efficiency than samples that have been spray-dried [34]. The food industry’s use of freeze-drying technology for encapsulation makes it possible to produce dry powder on a large scale and develop it as a finished product that is easy to store [35]. Thus, in this investigation, CCE microcapsules were made via the freeze-drying process.

Using EE as the index in the screening wall material experiment, the PPMD complex was determined to be the encapsulation CCE wall materials. Among them, MD, a carbohydrate material, is frequently utilized as an encapsulation wall material because of its superior functional qualities, which include excellent water solubility and resistance to a range of processing conditions. Currently, a small number of plant proteins, including those from soybeans and wheat, are the main focus of plant-based protein research [36]. As a plant-based protein, PP shares many physicochemical characteristics with SPI and is hypoallergenic. Customers with wheat or soy allergies may be able to choose from new options [17], which would make them viable substitutes for SPI [37]. When protein and carbohydrates are utilized as wall components, the protein is typically attached to the polysaccharide [38], which can work in concert to improve the antioxidant capacity and stabilize the active ingredient. The varying wall materials utilized in the bonding process result in variations in particle size, PDI, and zeta potential. While fewer CCEs were exposed on the microcapsule surface, PP-CCE, MD-CCE, and PPMD-CCE displayed greater brightness than free CCE. This is in line with the findings of Zhao et al. [13], which provide strong evidence for the high EE of PPMD-CCE.

A single-factor experiment determines the range of parameters studied after the preexperiment. There may be two reasons why the final process parameter EE is the highest. Tang et al. [39] reported that the folding of PP increases with increasing heating intensity, which may be more conducive to the combination of PP with MD and CCE. According to Kim et al. [40], adding MD improved EE and increased protein amphiphilicity and the β-sheet structure. Therefore, the best preparation parameters for CCE microcapsules, which are determined by single-factor tests and response surface tests, can increase the EE.

The better DPPH and ABTS radical scavenging activities of PPMD-CCE were demonstrated via in vitro antioxidant tests. Specifically, at all concentrations between 20 μg/mL and 100 μg/mL, PPMD-CCE had the highest ABTS free radical scavenging rate. This occurrence provides strong evidence that flavonoids have superior antioxidant potential due to the synergistic impact of polysaccharides and proteins [41]. Similar to the results obtained by Nallamuthu [27], the scavenging rates of DPPH free radicals were concentration dependent. Nallamuthu reported that the DPPH clearance rate of encapsulated naringin was 13.22% greater than that of free naringin. In this study, the clearance rate of the DPPH free radical in the encapsulated form increased by 107.35% compared with that in the free form. There are two reasons for this. The wall material is the first consideration, and research has indicated that both PP and MD are capable of acting as antioxidants. According to Febrianta et al.’s study [42], employing MD as a wall material can result in increased antioxidant activity. After glucosyl hesperidin was encased in a variety of plant-based proteins, Kopjar et al. [43] reported that PP particles had the highest EE and the best antioxidant activity. The structure of the microcapsule is the second factor. The manufactured CCE microcapsules have a higher BD, and the less air present, the better, preventing microcapsule oxidation, according to the experimental results of this study. In addition, the amorphous and flaky architectures of the three microcapsules can successfully shield the active ingredient from the damaging effects of air and high temperatures [44].

The ability of the human body to absorb and make its metabolites bioavailable is largely responsible for the many health benefits of flavonoids [45]. Research has indicated that metabolites generated following the breakdown of the intestinal flora are primarily responsible for the absorption of flavonoids into human blood circulation [46]. Therefore, it must be released rapidly and in large quantities in the colon and be as little influenced by stomach acids as possible to ensure its bioavailability. In this work, in vitro gastrointestinal digestion models were used to assess the bioavailability of three microcapsules. This experiment produced even better findings for PPMD-CCE. The artificial intestinal fluid had the highest release rate (85.89 ± 1.53%) and the lowest (19.75 ± 1.53%) in the artificial gastric fluid. In the simulated gastric juice digestion of anthocyanins by Wang et al. [47], the encapsulated complex also showed delayed release, with a release rate of only 53.5%. The above results are similar to those of Sun et al. [22]. During the 10 h simulated digestion process, the wall material protected the core material. The primary reason for this is still the efficient function of microencapsulation, which protects it from unfavorable gastrointestinal circumstances until it reaches the intended location [48]. Pepsin activity and the low pH of the artificial gastric juice digesting medium are responsible for the delayed release of CCE. In addition, MD can create a substantial insoluble layer as a wall material to shield the microcapsule from stomach juice digestion [49]. When used as a wall material, SPI can successfully postpone the release of active ingredients. Wu et al. [50] encapsulated blueberry anthocyanin extract using four different wall materials before deciding that SPI would have a greater protective impact and could help avoid diseases by promoting intestinal health. Trypsin and chymotrypsin, two pancreatic and intestinal proteases, are crucial for the breakdown of artificial intestinal fluid. Furthermore, the wall materials break due to the action of pancreatic enzymes and bile salts [51], releasing CCE. The environment also causes the protein’s carboxyl groups to ionize, which enhances the interaction between the polysaccharide and the aqueous medium. As a result, the CCE release rate has also increased [52].

The storage stability of CCE microcapsules can be increased in a variety of light and temperature environments. The first justification is the ability of microencapsulation to successfully shield the active ingredients inside. Moreover, the MC of desiccated microcapsules typically impacts their manufacturing, packing, and storage and is associated with their quality and stability [53]. When the MC content is between 4% and 10%, the MC generally has good storage stability [54]. The three microcapsules that we were able to collect for our study had MC values of 4.8900 ± 0.8253%–6.9467 ± 0.2250%. As a result, the three types of CCE microcapsules that have been created are resistant to mildew and deteriorate under MC, making them suitable for storage. Hygroscopicity also has a significant effect on product stability and storage. The term ‘hygroscopic property’ describes a product’s capacity to take in moisture from its environment. Low hygroscopicity might therefore prevent agglomeration and other microcapsule product occurrences [55], enhancing storage stability. Acacia gum was employed as an anthocyanin envelope wall material by Ramakrishnan et al. [56]. Storage stability tests at 4 °C and 25 °C revealed that wall materials with high moisture absorption facilitate relatively simple degradation of bioactive chemicals. Three types of microcapsules were found to have low hygroscopicity in our investigation; the lowest hygroscopicity, PPMD-CCE, was only 0.4818 ± 0.0010%. This could be explained by the structural alterations caused by the composite use of the two wall components. Moreover, MD is not very hygroscopic. Hygroscopicity can be substantially reduced during encapsulation by using or adding MD as a wall material [57]. Consequently, the product’s storage stability is better in microcapsules with MD as the wall material.

## 5. Conclusions

Overall, PPMD has the highest encapsulation rate for CCE among the studied wall materials, achieving an encapsulation rate of 84.44 ± 0.34% after optimizing the microcapsule preparation process, and the encapsulation system is stable. SEM, FTIR, and XRD images all indicate that the CCE has been successfully encapsulated within the wall material, ensuring its naturalness and integrity. The thermal stability of the CCE microcapsules was also confirmed via DSC. Furthermore, the encapsulation of CCE increased its antioxidant capacity, which was positively correlated with concentration, with PPMD-CCE demonstrating the best antioxidant potential. Additionally, PPMD-CCE exhibited the highest bioavailability in in vitro gastrointestinal digestion experiments, effectively protecting CCE from being released in gastric fluid while allowing for rapid and substantial release in intestinal fluid, thereby increasing its bioactivity. After being stored for 45 days under different temperatures and light conditions, encapsulation contributes to the stability of the CCE, which is attributed to the physicochemical characteristics of the CCE microcapsules, such as their low moisture content and low hygroscopicity, positively impacting their storage stability. In summary, plant-based proteins can be effectively utilized to encapsulate flavonoid compounds, resulting in excellent functional characteristics, whereas PPMD serves as an innovative and safe green composite wall material for encapsulating CCEs, laying a foundation for the future industrial application of CCEs.

## Figures and Tables

**Figure 1 foods-13-03096-f001:**
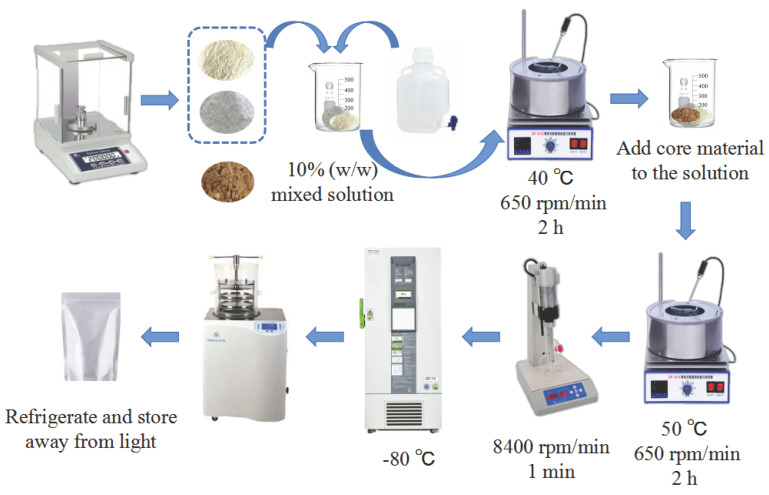
Flowchart of preparation of CCE microcapsules.

**Figure 2 foods-13-03096-f002:**
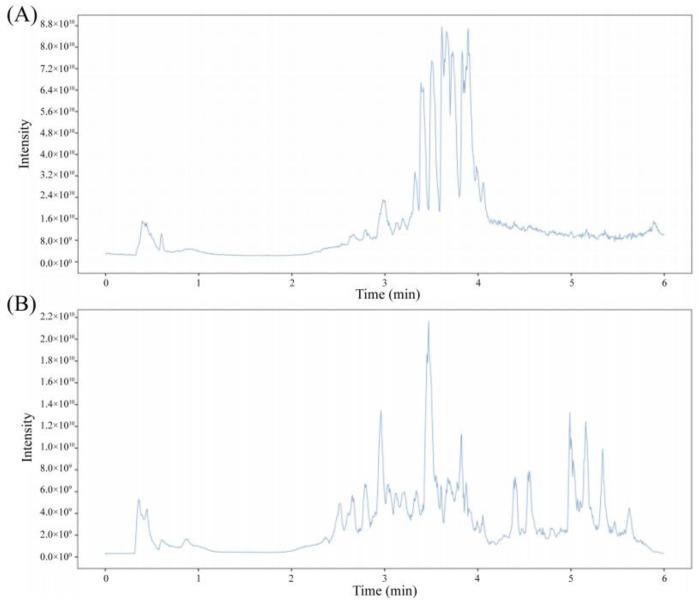
Positive and negative ion mode detection using UHPLC-OE-MS TIC chart for the CCE example ion modes: (**A**) positive; (**B**) negative.

**Figure 3 foods-13-03096-f003:**
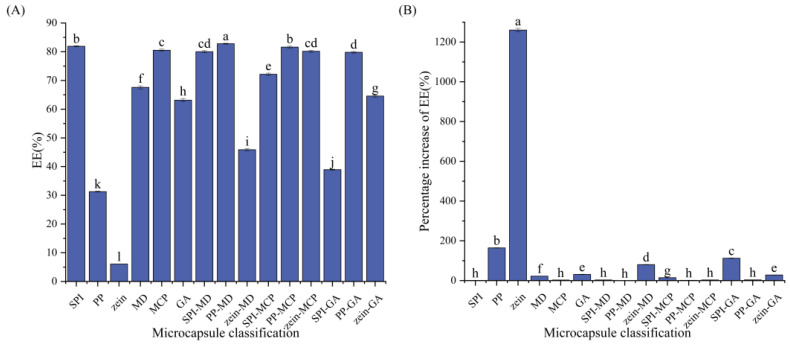
(**A**) EE of CCE microcapsules prepared by different wall materials. (**B**) The percentage increase in EE was compared between the best EE and the CCE microcapsules prepared with other wall materials. Superscript letters (a–l) denote significant (*p* < 0.05) difference in the same column.

**Figure 4 foods-13-03096-f004:**
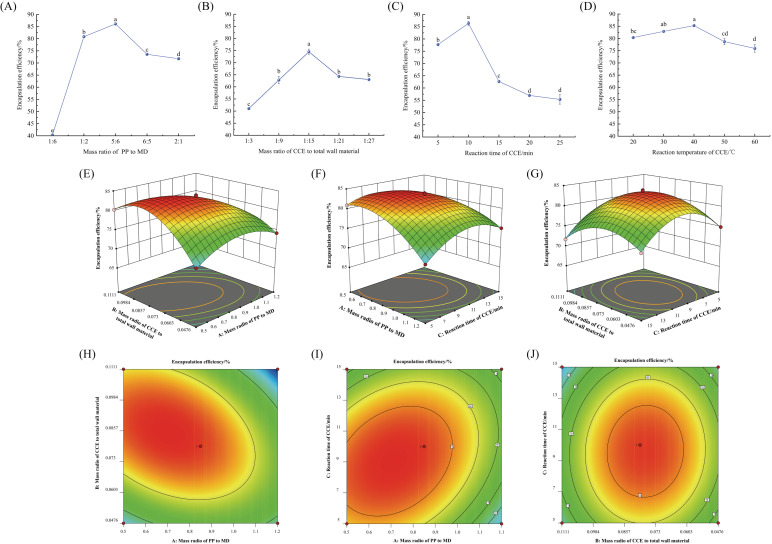
Optimization of preparation technology of PPMD-CCE: (**A**–**D**) single factor experiment; (**E**–**J**) the optimization of response surface methodology. Different superscript letters in the graph indicate significant differences between means (*p* < 0.05).

**Figure 5 foods-13-03096-f005:**
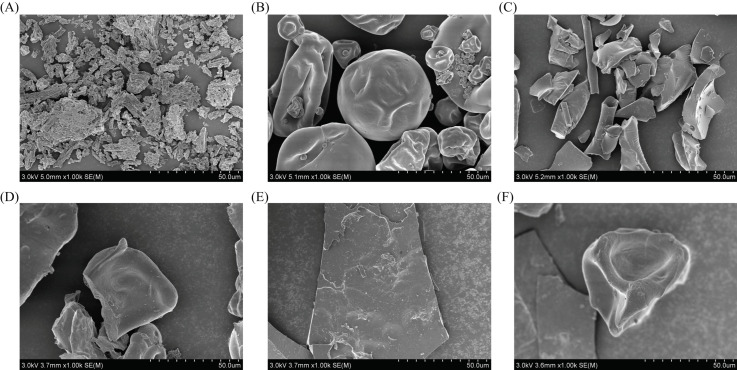
SEM image: (**A**) CCE; (**B**) PP; (**C**) MD; (**D**) PP-CCE; (**E**) MD-CCE; (**F**) PPMD-CCE.

**Figure 6 foods-13-03096-f006:**
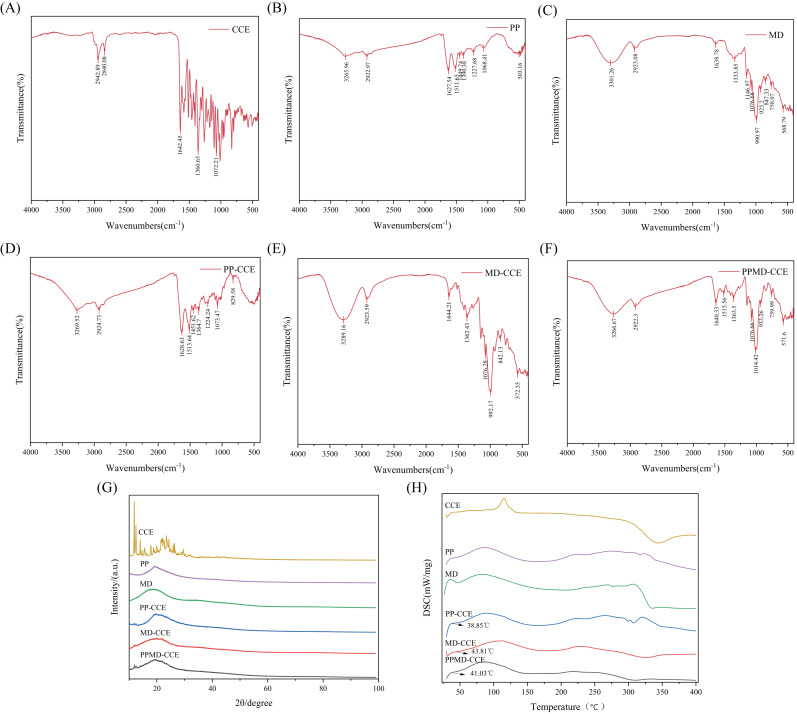
Characterization of CCE and CCE microcapsules: (**A**–**F**) FTIR spectra of CCE, PP, MD, PP-CCE, MD-CCE, PPMD-CCE; (**G**) XRD spectra of CCE, PP, MD, PP-CCE, MD-CCE, PPMD-CCE; (**H**) DSC analysis of CCE, PP, MD, PP-CCE, MD-CCE, PPMD-CCE.

**Figure 7 foods-13-03096-f007:**
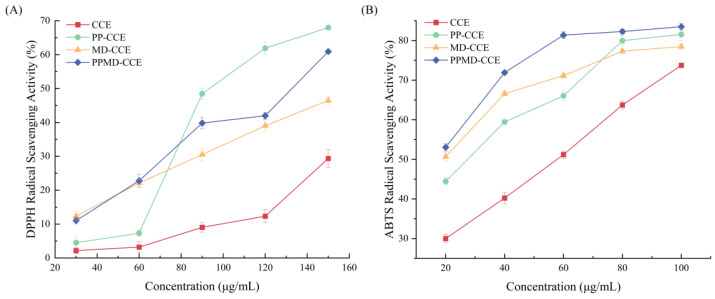
Antioxidant activities of CCE, PP-CCE, MD-CCE, PPMD-CCE. (**A**) DPPH radical scavenging activity; (**B**) ABTS radical scavenging activity.

**Figure 8 foods-13-03096-f008:**
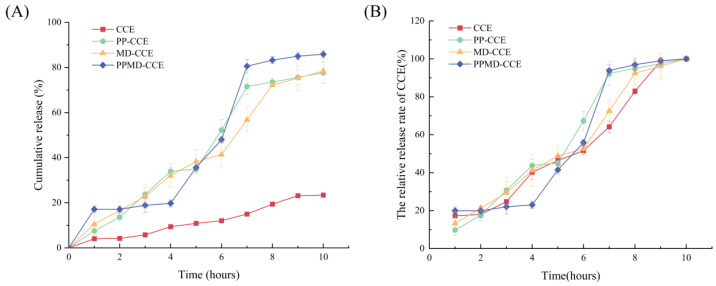
In vitro release profile of CCE, PP-CCE, MD-CCE, PPMD-CCE: (**A**) cumulative release; (**B**) the relative release rate of CCE.

**Figure 9 foods-13-03096-f009:**
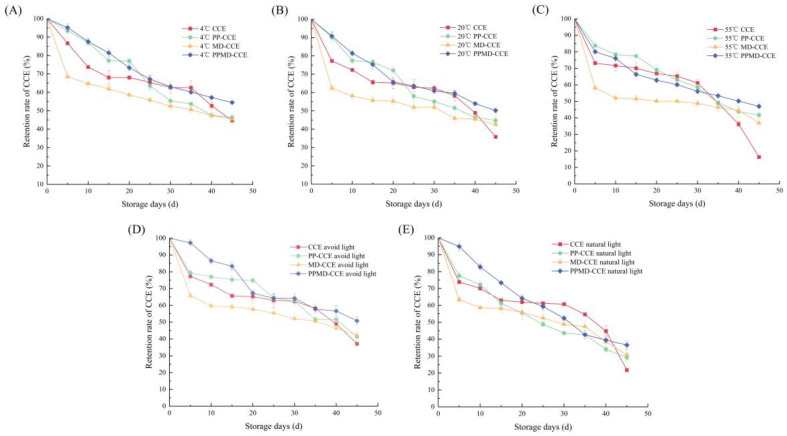
Retention rate of CCE stored under different conditions for 45 days: (**A**) storage at 4 °C; (**B**) storage at 20 °C; (**C**) storage at 55 °C; (**D**) storage avoiding light; (**E**) storage under natural light.

**Table 1 foods-13-03096-t001:** Factors and levels of response surface experiment for PPMD-CCE.

Factor	Coding Level		
	−1	0	1
Mass ratio of PP to MD (A)	1:2	5:6	6:5
Mass ratio of CCE to total wall material (B)	1:9	1:15	1:21
CCE response time (C)	5 min	10 min	15 min

**Table 2 foods-13-03096-t002:** UHPLC-OE-MS non-target metabolomics analysis of the content and percentage of the top 10 flavonoids in CCE.

Pound Name	CAS	RT/min	*m*/*z*	Peak Area	Proportion of Peak Area in Flavonoids/%
Hexamethylquercetagetin	1251-84-9	219.7	403.1383	92,268,427,012	79.7813
3′-Hydroxy-7,8,4′,5′-tetramethoxyflavone	133342-98-0	213.4	359.1121	8,889,119,462	7.6861
Heptamethoxyflavone	1178-24-1	221	433.1487	2,596,011,951	2.2447
Hesperetin 7-neohesperidoside	13241-33-3	177.4	611.1962	1,472,503,392	1.2732
Hesperidin	520-26-3	177.4	611.1962	1,472,503,392	1.2732
Narirutin	14259-46-2	176.2	579.1725	731,371,913.4	0.6324
Hesperetin 7-O-glucoside	31712-49-9	182.3	463.125	674,188,098.5	0.5829
Pectolinarigenin	520-12-7	182.1	315.0859	628,925,109.1	0.5438
Isosakuranetin		231.3	285.0769	458,444,185.7	0.3964
Citromitin		212.6	405.154	433,063,308.9	0.3745

**Table 3 foods-13-03096-t003:** Effects of different wall materials on EE% and 95% confidence interval (95% CI) of CCE microcapsules.

Wall Material	EE/%	Percentage Increase in EE/%	95% CI
SPI	81.90 ± 0.17 ^b^	1.080 ± 0.28 ^h^	[81.49, 82.32]
PP	31.29 ± 0.15 ^k^	164.55 ± 1.24 ^b^	[30.92, 31.67]
zein	6.09 ± 0.03 ^l^	1260.16 ± 8.43 ^a^	[6.02, 6.15]
MD	67.59 ± 0.55 ^f^	22.49 ± 0.85 ^f^	[66.23, 68.96]
MCP	80.54 ± 0.38 ^c^	2.79 ± 0.32 ^h^	[79.60, 81.47]
GA	63.14 ± 0.60 ^h^	31.12 ± 1.29 ^e^	[61.66, 64.62]
SPI-MD	80.02 ± 0.36 ^cd^	3.46 ± 0.38 ^h^	[79.13, 80.90]
PP-MD	82.79 ± 0.15 ^a^	control group	[82.41, 83.17]
zein-MD	45.89 ± 0.38 ^i^	80.41 ± 1.27 ^d^	[44.95, 46.84]
SPI-MCP	72.16 ± 0.41 ^e^	14.73 ± 0.68 ^g^	[71.13, 73.19]
PP-MCP	81.62 ± 0.36 ^b^	1.43 ± 0.55 ^h^	[80.73, 82.52]
zein-MCP	80.17 ± 0.38 ^cd^	3.26 ± 0.68 ^h^	[79.22, 81.13]
SPI-GA	38.93 ± 0.20 ^j^	112.68 ± 1.41 ^c^	[38.42, 39.43]
PP-GA	79.81 ± 0.26 ^d^	3.73 ± 0.51 ^h^	[79.17, 80.45]
zein-GA	64.57 ± 0.39 ^g^	28.23 ± 0.99 ^e^	[63.60, 65.52]

Superscript letters (a–l) denote significant (*p* < 0.05) difference in the same column.

**Table 4 foods-13-03096-t004:** Design and results of response surface experiment for PPMD-CCE.

Run	Factor A (Mass Ratio of PP to MD)	Factor B (Mass Ratio of CCE to Total Wall Material)	Factor C (CCE Response Time)	EE/%
1	0	1	1	71.72
2	−1	0	1	75.36
3	0	0	0	83.95
4	0	0	0	83.29
5	1	1	0	74.28
6	0	1	−1	73.68
7	1	0	−1	72.21
8	−1	1	0	72.76
9	0	0	0	83.82
10	0	−1	−1	74.97
11	−1	−1	0	80.29
12	0	0	0	82.27
13	−1	0	−1	80.95
14	0	−1	1	74.39
15	1	0	1	75.09
16	0	0	0	83.21
17	1	−1	0	69.22

**Table 5 foods-13-03096-t005:** Analysis of variance of regression model of PPMD-CCE.

Source	Sum of Squares	Degrees of Freedom	Mean Square	F-Value	*p*-Value	Significant
Model	389.73	9	43.30	145.78	<0.0001	significant
A	43.06	1	43.06	144.96	<0.0001	****
B	5.17	1	5.17	17.40	0.0042	**
C	3.45	1	3.45	11.60	0.0114	*
AB	39.63	1	39.63	133.41	<0.0001	****
AC	17.94	1	17.94	60.38	0.0001	****
BC	0.4761	1	0.4761	1.60	0.2460	
A^2^	50.96	1	50.96	171.57	<0.0001	****
B^2^	136.39	1	136.39	459.18	<0.0001	****
C^2^	64.92	1	64.92	218.54	<0.0001	****
Residual	2.08	7	0.2970			
Lack of Fit	0.3176	3	0.1059	0.2404	0.8644	
Pure Error	1.76	4	0.4404			
Cor Total	391.81	16				

* for *p* ≤ 0.05, ** for *p* ≤ 0.01, **** for *p* ≤ 0.0001.

**Table 6 foods-13-03096-t006:** The particle size and zeta potential of CCE microcapsules.

Formulation	Particle Size/μm	PDI	Zeta Potential/mV
PP-CCE	12.2333 ± 0.0929 ^a^	0.5371 ± 0.0133 ^b^	−29.0867 ± 0.6601 ^a^
MD-CCE	9.9567 ± 0.0681 ^c^	0.4583 ± 0.0220 ^c^	−52.2267 ± 0.3403 ^b^
PPMD-CCE	10.2733 ± 0.1966 ^b^	0.5877 ± 0.0216 ^a^	−55.1167 ± 0.7333 ^c^

Superscript letters (a–c) denote significant (*p* < 0.05) difference in the same column.

**Table 7 foods-13-03096-t007:** The chromaticity of CCE and CCE microcapsules.

Formulation	L*	a*	b*
CCE	53.38 ± 0.02 ^d^	4.72 ± 0.04 ^a^	18.39 ± 0.05 ^d^
PP-CCE	58.10 ± 0.02 ^c^	4.77 ± 0.03 ^a^	21.95 ± 0.03 ^a^
MD-CCE	63.65 ± 0.02 ^b^	2.74 ± 0.03 ^b^	19.17 ± 0.03 ^b^
PPMD-CCE	64.02 ± 0.01 ^a^	2.54 ± 0.02 ^c^	18.86 ± 0.01 ^c^

Superscript letters (a–d) denote significant (*p* < 0.05) difference in the same column.

**Table 8 foods-13-03096-t008:** Physical and chemical properties of CCE microcapsules.

Formulation	MC/%	WSI/%	WAI/%	BD/g/cm^−3^	Liquidity/°	Hygroscopicity/%
PP-CCE	5.6167 ± 0.6167 ^b^	14.80 ± 0.0800 ^c^	4.52 ± 0.0794 ^a^	0.3896 ± 0.0043 ^a^	29.4784 ± 0.6910 ^b^	15.4767 ± 0.0252 ^a^
MD-CCE	4.8900 ± 0.8253 ^b^	86.48 ± 0.7110 ^a^	0.28 ± 0.0265 ^c^	0.3822 ± 0.0042 ^a^	39.6437 ± 0.7148 ^a^	15.0533 ± 0.0306 ^b^
PPMD-CCE	6.9467 ± 0.2250 ^a^	49.02 ± 0.0640 ^b^	2.06 ± 0.0300 ^b^	0.3261 ± 0.0031 ^b^	38.3254 ± 0.5791 ^a^	15.0900 ± 0.0300 ^b^

Superscript letters (a–c) denote significant (*p* < 0.05) difference in the same column.

## Data Availability

The original contributions presented in the study are included in the article, further inquiries can be directed to the corresponding authors.

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
