# Peer review of "Microcapsule Preparation and Properties of Flavonoid Extract from Immature Citrus reticulata ‘Chachiensis’ Peel"

_foods, 2024, doi:10.3390/foods13193096_

Round 1

Reviewer 1 Report

Comments and Suggestions for Authors

Study on microcapsule preparation and properties of Citrus reticulata 'Chachiensis' peel flavonoid extract

The authors present the encapsulation of flavonoids using various encapsulating agents to improve stability, antioxidant capacity, and in vitro digestibility.

Improve the wording and correct writing of both units and formulas and writing style, as there are so many editing errors that it is difficult to read the manuscript.

Remove the word ‘study’ from the title because it is understandable.

Major revisions:

2.4.1

The authors repeat the methodology of section 2.3.

2.7.2

With what reference was the equipment calibrated?

What does b% mean?

2.7.4.

Why the water absorption index was not also calculated?

2.7.5

“The powder inside the microcapsules organically sank after repeated vibrations until the sample could no longer be squeezed”… How many vibrations?

2.7.8

Specify the magnifications performed.

2.7.11

What was the amount of milligrams used for the DSC tests?

2.9

What is the composition of the synthetic physiological fluids, and how were they prepared?

3.3.2

In the Box-Bhenken model, the authors must write in the tables the experimental values ​​to compare with those predicted by the model (Table 4).

3.4.1

“The distribution of the three microcapsules was uniform”… Why do the authors claim to have a homogeneous PDI?

3.5.2

In the characterization graphs, it is preferable to separate them. For the infrared spectra, it is necessary to include the wave number the authors refer to in the text.

3.5.4

When authors discussing the glass transition temperature, this fact cannot be seen in the graph. Discuss further or insert graphs that demonstrate the phenomenon.

3.6

Figure 4 does not show the error bars.

Minor revisions:

Throughout the document, italicize scientific names.

Write subscripts chemical formulas.

Authors should write in the methodology the brands, models, and countries of the equipment used appropriately.

Improve overall editing.

Comments on the Quality of English Language

Language is not well spelled, and scientific writing is not addressed.

Author Response

Dear reviewer,
We are very grateful for your suggestions, which are very valuable and constructive for our manuscript. We tried our best to finish the revision within ten days.Now the point to point response in the form of a document to upload. Major revisions have been marked in the latest manuscript using yellow highlighting. Thank you again for your dedication to your work.

Reviewer 2 Report

Comments and Suggestions for Authors

The manuscript presents a detailed study on the microencapsulation of flavonoid extracts from Citrus reticulata 'Chachiensis' using various wall materials, with a focus on plant-based proteins and carbohydrates. The study evaluates the encapsulation efficiency, physicochemical properties, antioxidant capacity, gastrointestinal release, and storage stability of the microcapsules. The study is well-structured, with a clear methodology and a comprehensive evaluation of the microcapsules' properties. However, I have a few recommendations:

Include more detailed explanations and justifications for the selected parameters in the encapsulation process, such as the choice of temperature, reaction time, and mass ratios. Additionally, consider adding a flowchart or schematic to visually represent the encapsulation process. The manuscript presents data with statistical significance, but it would be beneficial to include more comparative analysis, especially when discussing the efficiency of different wall materials. For example, while the manuscript states that PPMD-CCE has the highest encapsulation efficiency, a more detailed comparison with other materials (e.g., discussing percentage improvements) would provide a clearer picture of the relative performance. Incorporate confidence intervals or effect sizes alongside the p-values in the results section.

Author Response

Dear reviewer,
We are very grateful for your suggestions, which are very valuable and constructive for our manuscript. We tried our best to finish the revision within ten days. Now the point to point response in the form of a document to upload. Major revisions have been marked in the latest manuscript using yellow highlighting. Thank you again for your dedication to your work.

Reviewer 3 Report

Comments and Suggestions for Authors

The topic is really interesting,and authors used different characterization techniques to evaluate formation and stability of microcapsules with citrus extract. Few minor points need to be improved to be ready for publishing:

a) revise English through manuscript-there are lot of spelling mistakes

b) authors should include comparison of antioxidant activity, gastro activity, light and thermal stability of their microcapsules with same and/or similar microcapsules properties in literature.

Comments on the Quality of English Language

Minor english editing required

Author Response

(The authors gave the same response as above.)

Round 2

Reviewer 1 Report

Comments and Suggestions for Authors

Authors made some of the corrections suggested.

In table 4, it should be specified what the factors A, B and C are in the tittle.

For the color in section 7.4.2, the observation was about mentioning what the authors used as a blank for calibration (a whiteboard), not specifying the process of the equipment's use.

Section 2.4.1 for flavonoid content determination was corrected, but the language should be checked.

Section 2.2 is not aligned with the rest of the document.

The composition of synthetic gastric juice was not specified.

Comments on the Quality of English Language

Language should be checked for minor corrections.

Author Response

Dear reviewer:
Thank you again for your valuable and constructive advice. These suggestions are undoubtedly beneficial to our manuscript. We have revised the problem you raised.Point to point response in the form of word document upload in the attachment on this site. Major revisions are highlighted in blue in the manuscript. Thanks again for all your hard work.
